# Review of Urban Access Regulations from the Sustainability Viewpoint

Yunpeng Ma and Ferenc Mészáros *

Department of Transport Technology and Economics, Faculty of Transportation Engineering and Vehicle Engineering, Budapest University of Technology and Economics, Műegyetem rkp. 3., H-1111 Budapest, Hungary; mayunpeng@edu.bme.hu
* Correspondence: meszaros.ferenc@kjk.bme.hu

**Abstract:** This article reviewed the urban vehicle access control policies derived from disparate spatiotemporal dimensions that aim to eliminate the negative externalities of traffic caused by urbanization. Urban access regulations are important tools often required to achieve the sustainable mobility vision of cities. Employing a systematic literature review methodology, this review summarized and analyzed various urban access control policies to enlighten policymakers and future scientific research. The results indicate that combinations of multiple-dimensional restriction policies (including inter-policy and intra-policy) have more significant effects than implementing a single policy. Classified according to their objectives, control policies were discussed in terms of their benefits and limitations. The authors are inspired to propose and describe five paradoxes of urban access control policies.

**Keywords:** sustainable mobility; urban mobility; traffic restriction; vehicle access control; traffic zone

## 1. Introduction

Half of the population in the world lives in urban areas, and the urban population is projected to reach 6.7 billion by 2050 [1]. Many dilemmas are associated with rapid and massive urbanization, such as air pollution and transport congestion [2]. Among these problems, the negative externality of urban transport inhibits sustainable urban development. Besides the abovementioned problems, externalities are also observed in noise, emissions, and traffic accidents [3].

To eliminate the negative effects of urbanization, there is a consensus on sustainable development worldwide. Sustainable transport has been widely applied as a derivative of sustainable development [4]. To accomplish this vision, governments generally implement two types of policies: motivation policies and regulation policies. For instance, there are tax benefit policies for electric vehicles in the Netherlands [5], public transport subsidies in Colombia [6], Park & Ride (P + R) parking lot expansion in Hungary [7], grants for the installation of Electric Vehicle (EV) charging points in the UK [8], and electricity subsidies for EVs in China [9].

On the other hand, various vehicle access regulations within urban areas have been proposed and implemented; these regulations are the main research fields of this study. Considerable research has evaluated the results of extant implemented access control policies based on city case studies. This study aims to systematically review and analyze the latest five years of published research and explore the valuable experiences and shortcomings of these methods in achieving sustainable transportation.

Following the background introduction, a paragraph will describe the previous related works and research gaps in the field. Then follows the Methodology section, which elaborates on the entire literature selection process. Next, the effects of these regulations on the environment, society, and economy will be discussed in detail in the Results section. After the summary, the authors found five paradoxes in urban access control policies, which

will be disclosed in the Discussion. Finally, the Conclusion chapter will state the limitations of this study and further works.

## 2. Related Works

Through the inspection of the review articles published during 2019–2023 (in English), the authors found works only by Moretti et al., who focused on compiling the pros and cons of urban access control policies, which are limited in Europe [10]. Given that there were no specific review articles on urban access regulations related to all three sustainability aspects published during the last five years, most of the relevant research concentrated on evaluating the performance of policies in a single dimension from the three. Therefore, the authors extracted some common negative effects from the actual implementation and emphasized them as the paradoxes of urban access control policies in this paper. Such a systematic review can help policymakers and stakeholders optimize the achievement of policies as well as avoid external costs arising from the policies themselves, which can be the research gaps in this field and, therefore, the novelty of this paper. Sustainability, as a hotspot, usually has multi-dimensional meanings from one paper to another. The authors used the following definitions for sustainability's dimensions [11]:

- Environment: Urban transport policies related to environmental pollution (air, water, land), climate change, energy consumption and emission, and any sensory factors that make people uncomfortable (e.g., noise, cityscape destruction, pungent exhaust, etc.);
- Society: Policies aiming to improve social equity, safety, acceptance, reputation, civic engagement, and utilization of public road resources.
- Economics: Cost (internal, external, and operating), profit, implications for urban productivity, and investment of urban transport policies.

## 3. Materials and Methods

This study mainly explored the practicability and appropriateness of urban access control policies. Thus, based on related research [12], the authors considered the systematic literature review (SLR) the best-fitting method. Moreover, compared with a scoping review, an SLR could generate a greater effect on policymaking, which fits the outlook of this article [13]. In detail, the systematic literature review method was used for this review article for three reasons:

- the high degree of transparency of the chosen literature [14];
- the clear and replicable review process [15]; and
- the great generalization of urban access regulations with different spatial-temporal dimensions.

Figure 1 shows the review process steps, allowing the reviewers and readers to inspect the logic of the method steps easily. Furthermore, this methodology can be replicated in future studies, ensuring consistency in approach. The literature was collected between 18 September and 15 November 2023.

As stated in the introduction, this review focuses on access regulations within urban areas. Therefore, regulation policies implemented in non-urban areas, large regions, and inter-regional areas are not considered. Although some urban transport policies also aim to improve traffic efficiency and reduce emissions, such as purchasing restrictions on plate numbers and vehicles, these policies are not discussed due to their divergence from the focus on access control.

Table 1 lists the 6 criteria for selecting and excluding the identified studies for this review.

Based on these above criteria, 89 articles were selected (Table S1) by filtering from a total of 4755 articles (Figure 2).

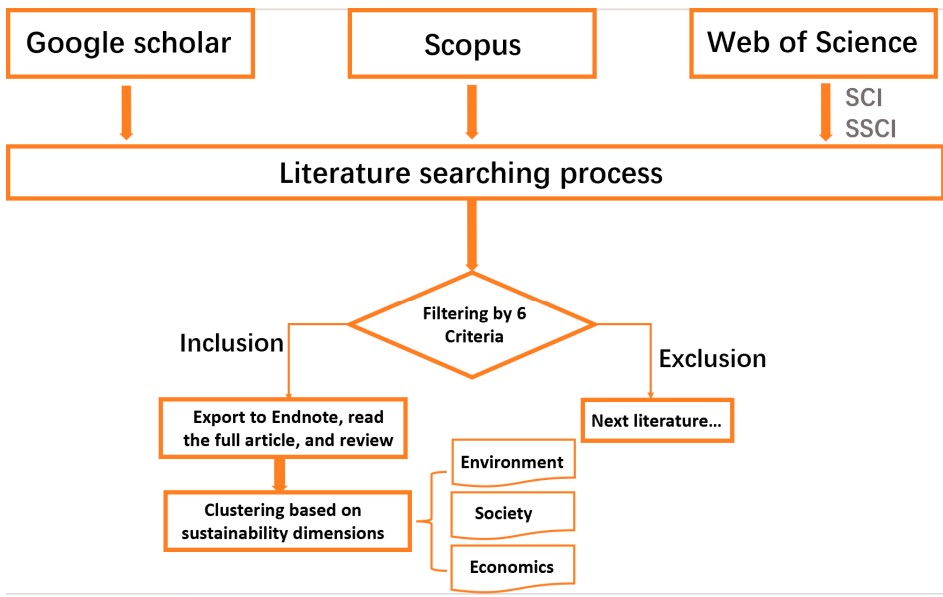

**Figure 1.** Flowchart of the systematic literature review (source: own edition).

**Table 1.** The literature inclusion and exclusion criteria (source: own edition).

| | Criteria | Decision |
|---|---|---|
| 1 | The study was published in the last five years (2019–2023) and in English | Inclusion |
| 2 | The title and abstract cover urban access regulation policies and address at least one of the three dimensions of sustainability | Inclusion |
| 3 | The literature lapsed or not included within the affiliations access rights [1] | Exclusion |
| 4 | The same study has appeared in another database | Exclusion |
| 5 | Articles that only present new technologies without discussing policy | Exclusion |
| 6 | The study was published before 2019 but has been previously cited by more than a single paper | Inclusion |

[1] The literature is not open-accessed, and the affiliation does not have access rights. Very few literature links are invalid.

To cover more related research, the authors used multiple keywords, including "urban", "vehicle", "traffic", "access", and "transport". Due to the rapid development of vehicle and energy-related technology, the literature is limited to English-language journals and conference papers published in the last five years because policies vary with time. A typical example is the emission standard of conventional fueled vehicles and EVs. However, some literature published earlier, cited by the latest papers, is also valuable. For the search databases, Scopus and Web of Science, renowned scientific databases utilized globally, served as the primary search databases. Additionally, Google Scholar, with its extensive coverage of internet sources, was also used as a supplement. The list of titles and abstracts was the second level of filter used to screen and select relevant papers.

The authors investigated an SLR-type research structure [15] and applied it to this research. Table 2 shows the details of the 2nd filtering (Criteria 2) process.

Endnote (X9) was used to manage and make notes on the literature. According to the aim of this article, the selected literature is categorized into three groups (environment, society, and economics) based on three aspects of sustainability.

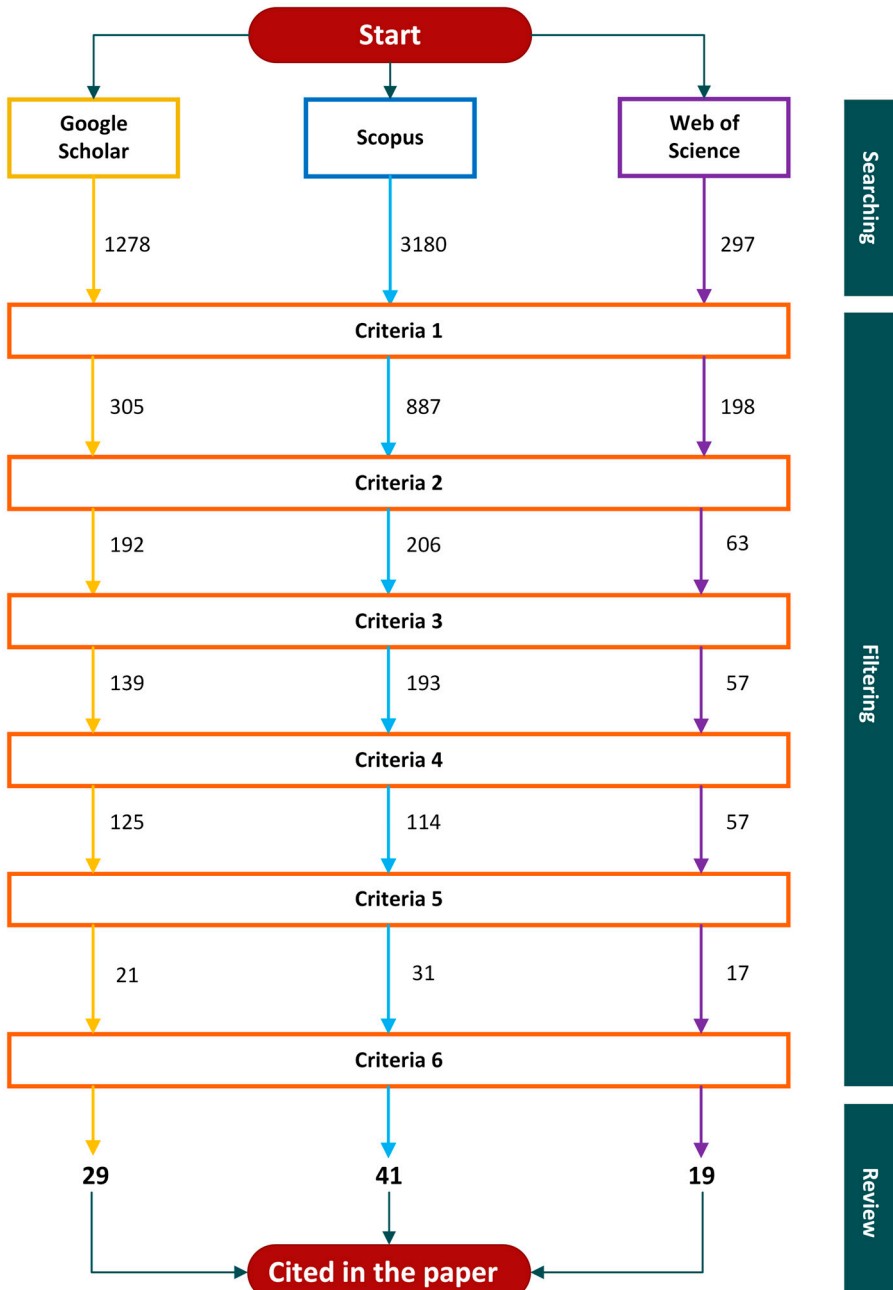

**Figure 2.** Flowchart of the literature exclusion (source: own edition).

**Table 2.** The literature search process.

| Database | Main Search Terms [1] | Secondary Search Terms/Disciplines | No. of Papers after 1st Criteria |
|---|---|---|---|
| Google Scholar | "urban" | "access control" | 3 |
| | | "access restriction" | 3 |
| | | "traffic restriction" | 58 |
| | | "transport regulation" | 17 |
| | | "traffic zone" | 6 |
| | "vehicle" | "access control" | 168 |
| | | "access restriction" | 29 |
| | | "traffic restriction" | 17 |
| | | "transport regulation" | 0 |
| | | "traffic zone" | 4 |
| Subtotal | | | 305 |

**Table 2.** *Cont.*

| Database | Main Search Terms [1] | Secondary Search Terms/Disciplines | No. of Papers after 1st Criteria |
|---|---|---|---|
| Scopus | Advanced search [2] | Engineering | 717 |
| | | Social Sciences | 160 |
| | | Environmental Science | 119 |
| | | Energy | 66 |
| | | Economics | 16 |
| Subtotal [3] | | | 887 |
| Web of Science | Advanced search [4] | Transportation science technology | 106 |
| | | Environmental Science | 70 |
| | | Environmental Studies | 33 |
| | | Transportation | 30 |
| | | Green sustainable science technology | 28 |
| | | Economics | 14 |
| | | urban studies | 3 |
| | | Social Sciences | 2 |
| Subtotal [5] | | | 198 |

[1] Last five years of the literature. [2] TITLE-ABS-KEY ((urban OR vehicle) AND ("traffic restriction*" OR "transport regulation" OR "access control*" OR "access restriction*" OR "traffic zone")) AND (LIMIT-TO (SRCTYPE, "j") OR LIMIT-TO (SRCTYPE, "p")) AND (LIMIT-TO (LANGUAGE, "English")). [3] Some literature covers multiple areas. [4] TS = ((urban OR vehicle) AND ("traffic restriction*" OR "transport regulation" OR "access control*" OR "access restriction*" OR "traffic zone")). [5] Same as 3.

## 4. Results

### 4.1. Overview

The word "sustainability" has been under public scrutiny since the 1980s; concerns regarding the adverse impacts of transportation emerged, including:

- air pollution;
- climate change;
- traffic congestion (including fuel cost, extra pollution, time waste, and cargo delays);
- road safety;
- traffic equality [16,17].

Urban access regulations have been implemented worldwide to achieve sustainable mobility rather than simply to restrict city traffic [18]. However, the latest related research has mainly focused on Asia and Europe (Figure 3). Wu et al. noted that American cities have a lower urban cluster density than cities in Europe and China [19]. More roads were expanded and built through urbanization, which attracted more vehicles (induced demand) and objectively promoted passenger car dependence and congestion [20]. Furthermore, densely populated and mixed cities in Asia and Europe present significant opportunities for implementing transportation regulations and transformations, such as biking and walking.

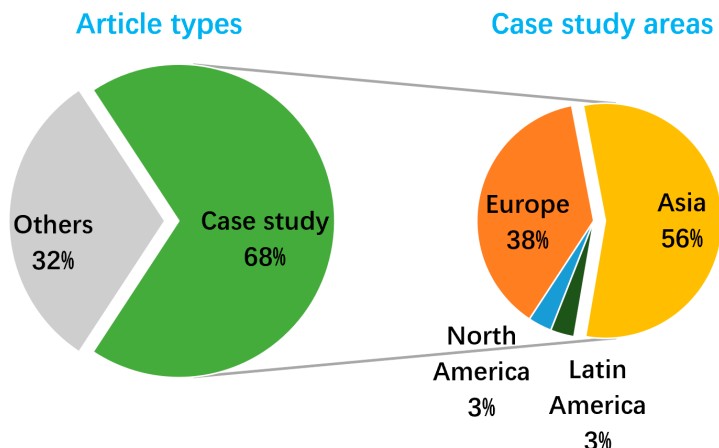

**Figure 3.** Composition of the explored literature and study areas (source: own edition).

Urban access control refers to a combination of Urban Vehicle Access Regulations (UVAR), Traffic Restriction Policy (TRP), Driving Restriction Policy (DRP), and Motor Vehicle Restriction (MVR) in different studies. According to the limited subjects, these policy tools can be classified into three types: time-based; vehicle-based; and spatial-based (Figure 4). Each identified restriction policy is discussed in the following chapters.

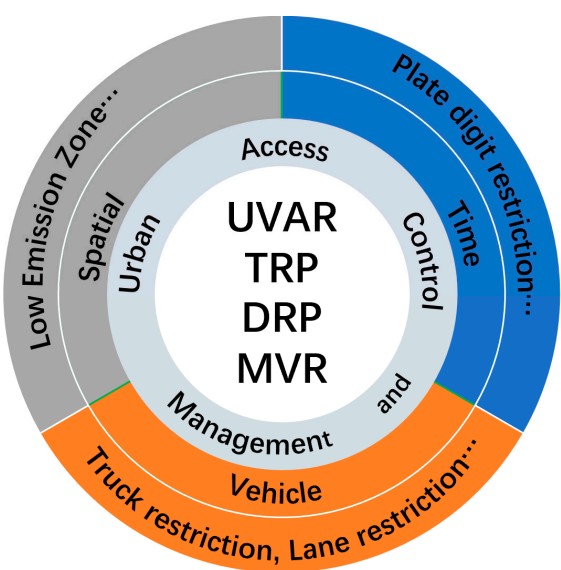

**Figure 4.** Urban access control classification (source: own edition).

Table 3 lists the specific urban access regulations discussed in this research by region.

**Table 3.** Specific classification of urban access regulations in this study.

| Region | Policies | Detailed Categories |
| --- | --- | --- |
| Europe | Speed limit<br>Low emission zone | 30 km/h |
| | Congestion charging | Daily<br>Entry times |
| | Parking Management (PM)<br>Truck Restriction Time Window (TRTW)<br>Zero emission zone [1]<br>Traffic-free areas around schools | |
| Asia | License plate digit restriction (workdays) | odd-and-even<br>one day per week |
| | Non-Local Vehicle Restriction Policy (NLVDRP) | peak hours<br>diesel vehicle |
| | Waste transportation trucks prohibition (during events)<br>Lane restriction [1]<br>Road closure (during events)<br>Road pricing<br>TRTW<br>Bridges restriction<br>TRP-carpool exemption: high occupancy vehicle [1] | |
| Latin America | License Plate Restriction Charging (LPRC) | Workdays<br>Saturday |
| North America | Speed limit in school areas | 30 km/h |

[1] They have not yet been put into practice but are rather a concept.

After analyzing the 89 selected papers, the works were categorized according to the three aspects of sustainability discussed above.

The problems generated by the negative externality of transport usually cannot be classified into a single aspect. For example, congestion creates a social nuisance for citizens while increasing emissions. In this paper, the specific problem categorization adopted the approach of Ogunkunbi and Meszaros, who constructed an assessment structure for UVAR from five aspects besides three aspects of sustainability. They also proposed technical and governance structures [21].

*4.2. Environment*

One of the visions of urban access regulations is to reduce air pollutants, such as Nitrogen Oxides (NOx), particulate matter, and Greenhouse Gases (GHGs). From this point of view, various environmental policies published by different countries are related to transport [22].

After the impressive results in Sweden in the 1990s, the Low Emission Zone (LEZ) became a popular policy instrument in Europe and has been considered across 315 European cities [23,24]. Every year, the EU pays between EUR 67 and 80 billion to address the indirect health costs associated with traffic-related air pollution, 75% of which are related to diesel vehicles. By implementing LEZ, these costs could be reduced by approximately 70% by 2030 [25]. However, diverse cities have different restriction levels according to the local environment. For example, in the LEZ in Madrid, vehicles with high emissions are not allowed to enter the area [26]. With this strict regulation and parking management, Madrid has achieved broadly successful outcomes in reducing emissions [27]. In this case, all the cities whose populations exceed 50,000 or have a population of more than 20,000 but with high levels of air pollution should establish LEZs before the end of 2023 [28]. Another study in North Macedonia indicated that such regulations (LEZ and PM) are effective at reducing the emissions generated by high-emission volume vehicles [29]. In Rome, some limited traffic zones had night bans for some types of vehicles, while some others required extra payment [30,31].

Nonetheless, the results of implemented LEZs are not always satisfactory. In Barcelona, despite the implementation of Low Emission Zones (LEZ) and speed limit policies, the air quality index continued to fall below the standards set by the European Union for air quality [32]. Research conducted in Rome revealed that concentrations of $NO_2$ and $PM_{10}$ exhibited reductions solely within the boundaries of the Low Emission Zone (LEZ), with decreases of 23% and 10%, respectively, while the overall city-wide levels remained unchanged [33]. Although the Low Emission Zone (LEZ) was deemed successful in downtown Madrid, an increase in traffic volume was noted at the perimeter of the LEZ, suggesting a displacement of traffic to other areas of the city. This phenomenon can be identified as an externality of the LEZ [34]. Similar spillover effects were also observed in other access regulations reviewed in this study.

Another widely used policy is vehicle license plate digit restriction, which first emerged in the 1980s [35]. According to Fransen et al., this policy was preferred by Asian (especially China) and Latin American countries (Mexico, Colombia, and Brazil) [36]. As illustrated in Figure 2, most of the case studies were from Asia during the last five years. Plate digit-based restrictions implemented in different large Chinese cities can be attributed to the extraordinary results of the policy imposed during the Beijing Olympics (2008) [37]. Vehicles are prohibited on urban roads based on the last plate digit (alternative odd or even number restriction). After a two-month restriction, the government kept this policy, but one day per week, and only constrains two numbers (from 0 to 9) per workday. Compared with the period before the Olympics, air pollution dropped by 21% under the one-day-per-week restriction [38]. From a medium-term perspective, such restrictions alleviate traffic congestion without causing spillover effects [39].

Generally, odd–even restrictions are implemented to mitigate emergency air pollution due to the extensive coverage of vehicles [40]. An example in China is Jinan, which enforced the odd–even restriction as a temporary policy to alleviate air pollution. The concentrations of CO and $PM_{10}$ were reduced by 46.6% and 33.1%, respectively [41]. The

effect of the one-day-per-week policy varies from place to place. Li et al. [42] determined that weekday traffic speed has increased by 6.3–9.8%, even more during peak hours. This result was good because more $NO_2$ is generated during peak [43]. Another study evaluated 20 Chinese cities that implemented plate digit restrictions (with 14 one-day-per-week and 6 odd and even), and only 3 cities had good performance in reducing winter smog and haze pollution. The reasons are likely to be the industrial structure of the city, public transportation development, and meteorological factors [44]. We found similar conclusions in other studies. Stathopoulos and Argyrakos stated that pollution on urban roads is a complex function of time, transport, and meteorological conditions [45]. In Rio de Janeiro, Brazil, the reduction in traffic volume did not induce a subtraction of contaminated particulate matter because of meteorological and industrial factors [46]. In Mexico City, plate digit restriction was even extended to Saturday. However, the results were not satisfying because the air quality did not improve because drivers did not shift to low-carbon transportation modes [35].

Similar access restriction schemes may yield diverse effects on air quality in different cities within the same country, as illustrated by a study comparing policy impacts between northern and southern cities in China [47]. Major pollutants ($PM_{2.5}$, $PM_{10}$, CO, $NO_2$, and $SO_2$) were greatly reduced in Chengdu, China, by expanding the one-day-per-week restriction area but were observed only in the regulated area [48]. There was a special case in Chongqing, China, where a one-day-per-week policy was used for bridge access control due to its unique geographical and hydrological environment. However, the improvement in air quality in only one district exacerbated the deterioration of air quality in seven other districts, which was a typical spillover effect. Since drivers are unfamiliar with the detour roads and increased itineraries, the private traffic was not diminished but was redistributed in the spatiotemporal dimension [49]. In the port city of Izmir in Turkey, despite implementing urban access control for trucks, the queues caused by peak terminal operations continued to generate high emissions [50].

The NLVDRP, also known as the Urban Core-focused License-plate-based Driving Restriction Policy (UCLDRP), was commonly implemented in China with the aim of regulating access to the primary urban areas by restricting vehicles with non-local license plates. For instance, approximately one-third of vehicles in Shanghai possess non-local license plates since they are much easier to obtain than local licenses. The restriction typically occurs during peak hours to improve air quality and ease congestion. Surprisingly, this approach proved ineffective ($NO_2$, $PM_{2.5}$, and $PM_{10}$ concentrations increased, and the traffic speed during the night peak only rose by 1.47%) [51,52]. The spatiotemporal spillover effect caused CO and $PM_{10}$ to increase (16% and 8%, respectively) at the border. Even a 10–20% increase in CO and $PM_{10}$ concentrations was observed before and after the restriction time [51]. Another Chinese megacity, Shenzhen, restricted access to non-local diesel vehicles. However, the effect was limited. The main reasons for emission are acceleration and deceleration due to intersection congestion [53].

In 2020, transport noise affected approximately 113 million citizens in the European Union [54]. Speed can be controlled in urban areas to reduce noise as it is a basic parameter in traffic noise levels. Research in Lodz, Poland, revealed that 9469 buildings could be released from the effect of traffic noise through a speed limit below 50 km/h [55]. In the Swiss city of Lausanne, a 30 km/h speed limit effectively reduces traffic noise and fatalities [56].

In addition to the above policies, the authors also found other case studies about access control to advance the environment, such as more draconian non-local access restrictions during public events (China International Import Expo) in Shanghai, which extended the restriction time to all day. The concentrations of $NO_X$ and CO, but not that of particulate matter, became significantly lower. Therefore, pollutant type and spatial allocation are critical for policy effectiveness [57]. Similarly, Beijing implemented non-local heavy diesel vehicle access control for all days during the Winter Olympic Games in 2022, which resulted in a 20–30% decline in pollutants [58]. One study in Oslo found that improving

parking prices is the most effective way to control air pollution [59]. Moreover, Yang et al. determined that the traffic regulations during the Beijing Olympic Games led to a 1.5–2.4 °C ground average temperature drop and an 820 km$^2$ heat range narrowing [60].

In summary, achieving environmental sustainability in urban areas requires a comprehensive strategy by urban authorities. Firstly, it is important to collect local real-life traffic data and integrate various policies, such as enhancing the quality of public transport (transitioning to electric public transport) and improving the quality of walking and biking [61–63]. Secondly, access control policies should not remain static, as their marginal benefits decrease with the adoption of renewable energy vehicles [64]. Thirdly, access control policies should be tailored to local circumstances rather than merely replicating others' implementations. Factors such as climate, geography, and the industrial structure of a city must be considered when determining control measures [65]. Lastly, new technologies are on the way. Gauna et al. have proposed an urban pollution protocol in Madrid, aiming to carry out traffic restrictions dynamically according to the level of pollution [66]. Wei et al. invented a special restriction exemption for High Occupancy Vehicles (HOVs) to compensate for traffic restriction policies sustainably [67]. Moreover, applying control policies helps reduce the need for citizens to buy and use vehicles [68].

### 4.3. Society

Congestion is widely recognized as the largest negative externality of urban transportation [69]. Congestion Charging (CC) and Road Pricing (RP) policies, albeit less frequently implemented than LEZ, have been deployed in Europe to promote sustainable urban transport due to their environmental and social benefits [30]. As the names suggest, drivers must pay when they access busy roads and areas, typically in city centers. CC and RP have significantly reduced air pollution in London and Stockholm. However, they share a common limitation—their effects are often limited to the charging area [70,71]. Compared with restriction, charging policies have lower social acceptance [30]. A critical aspect of exploring urban access regulations from a societal perspective is citizen acceptance of access control policies. A survey of 1300 residents in Beijing's proposed CC zones revealed that residents residing in densely populated areas near public transportation hubs and the city center were more supportive of CC [72]. In Athens, it was observed that males and young individuals were more likely to accept an annual road pricing policy [73]. Moreover, CC and RP have shown spillover effects (increased congestion at the peripheries) in case studies as well [30,71]. Zhang and Kockelman noted that CC is most effective in heavily congested cities and should be implemented in conjunction with other policies in less congested cities [74].

Parking-related issues are significant in urban areas because they contribute to 30–50% of the total traffic volume during peak hours [75]. There is usually a shortage of parking spaces in the Central Business District (CBD) because of high levels of urbanization, which worsens congestion and generates accidents and noise [76]. Worse, the shortage increases illegal parking along the street, which causes more ignorance of traffic statutes [77]. Parking management is commonly viewed as complementary to other policies. For instance, in Madrid's LEZ, parking fees are contingent on vehicle emission standards to incentivize low-emission vehicles [78]. Bhavsar et al. suggested avoiding the establishment of parking spaces near roundabouts [79]. A recent study proposed using Autonomous Vehicles (AVs) to transport passengers in the CBD and then park in an area where parking is cheaper [80]. However, as with the previous policy, these parking management approaches also exhibit spillover effects, with increased parking observed around the periphery of the policy coverage area [78].

Several studies have highlighted concerns that wealthier families could afford a second vehicle to avoid these restrictions [81–83]. This socio-economic disparity not only undermines the efficacy of regulatory policies but also provokes negative sentiments among citizens toward urban traffic control policies. Conventional solutions to address this emission reduction paradox often involve a combination strategy leveraging synergistic effects

and stakeholder involvement. The most frequently mentioned in the studies is the integration of access regulations with improvements in public transportation, biking, and walking, such as improving the comfort level of public transportation (e.g., safety and peak hour headways) and constructing more infrastructure for biking, thus increasing acceptance by citizens [81,83–85]. However, the shift of passengers from private vehicles to public transit during restriction periods may lead to overcrowding at popular stations, which is called the agglomeration effect [86]. Traffic authorities can mitigate this congestion by strategically managing passenger flow based on traffic regulations [87]. In addition to public transportation, micro-mobility options also require effective management. Electric scooters are an emerging form of public micro-mobility. However, issues such as haphazard parking and safety hazards have prompted considerable criticism of electric scooters in urban areas, as reported by Wallgren and colleagues [88]. Conversely, stakeholders, including government bodies, specialists, and citizens, are expected to participate in policymaking processes [89]. Shi et al. stated that most of the policy information was unidirectionally passed from the administration to the public [90]. Citizen engagement at the policymaking stage can help to better define the focused problem, reach a consensus, identify potential risks, and ultimately increase social acceptance [90–92]. Furthermore, drivers' psychological attributes are important contributors to the emission reduction paradox. High-income citizens tend to have higher levels of private vehicle dependency because it gives them psychological independence, a sense of ownership, and social status [93,94]. Moreover, wealthier people are capable of purchasing Battery Electric Vehicles (BEVs) or other alternative-fueled vehicles, which are often exempted from urban access control policies. For example, BEVs are exempted from urban traffic restrictions in China [95]. While transitioning from conventional vehicles to BEVs is currently a common policy objective, these exemptions for BEVs may not be sustainable in the long term. Ogunkunbi and Meszaros argued that phasing out subsidies for hybrids and BEVs will be necessary in the future [96].

Schools are a special case regarding reducing private transportation and access control. In Calabria, Italy, schools can attract more than 20% of the total population during peak hours in the city center [97]. A study in Beijing indicated that schools increase the probability of traffic congestion by 4.5% [98]. The authors found several proposals for access control related to schools:

- traffic-free zone [97];
- speed limit [99];
- stagger school schedules; CC in school areas [98].

In addition to exploring drivers' acceptance of the policy, attention has been drawn to violations. Two studies illustrated that non-local drivers, those who live far from city centers and those who usually drive during peak hours, are more likely to contravene the restrictions [100,101].

The social aspect of urban access control aims to create a more liveable city, diminish social inequities, and increase residents' satisfaction with the transportation environment. Implementing various policies, such as CC, parking management, and limited traffic zones, efficiently reduces private transport in urban areas [102]. It is imperative to promote the decoupling of private vehicles from social identity. Furthermore, enhancing the quality of public transportation, sidewalks, and bike lanes and introducing new transportation modes, such as shared mobility options, should be prioritized [103,104].

*4.4. Economics*

Compared with the other two aspects, the literature examining these policies from an economic perspective is relatively scarce. However, considering that social and environmental aspects also entail economic consequences in terms of social welfare and external costs, the authors add only the additional aspects to this section. Unlike the other two aspects, the primary objective of urban access regulations in economics is the internalization and reduction in external costs rather than generating economic profits. Nonetheless, there can still be economic benefits associated with urban charging policies, such as CC, RP,

and LEZ. All of these policies have been described above, but in addition, some are more economically flexible, such as the High-Occupancy Toll (HOT) lane. Derived from High-Occupancy Vehicle (HOV) lanes, HOT lanes allow users to choose whether to pay to bypass congested roads via HOV lanes, even if there is only one occupant in the vehicle [105]. The primary objectives of these charging policies are twofold: first, to supervise and regulate urban transportation, such as reducing highly polluting vehicles and congestion. The other is to finance public transport and infrastructure and provide subsidies to disadvantaged groups (e.g., concessionary bus tickets) for social equity. Nevertheless, all charging policies have both of these effects [106]. The fulfillment of these two objectives is the criterion by which the effectiveness of the policy is judged from the economic perspective. Singapore serves as an exemplary case in this regard. The government sustains an affordable public transport price using revenues from Electronic Road Pricing (ERP). A greater number of public transportation passengers and a wider public transportation network could prove the success of this approach [107]. Jun et al. demonstrated that implementing CC policies could lead to a reduction in congestion-related costs by EUR 192 million and generate a profit of EUR 1.3 billion in Beijing [108]. However, in Cali, Colombia, the LPRC was implemented, and while the annual income of the LPRC policy greatly increased, the contribution to public transport costs was less than 2% [109].

In addition to the financial revenue generated, urban access regulation policies provide some intangible benefits. For example, reduced traffic volume allows for increased efficiency and safety for drivers and cyclists alike [106].

Besides revenue, the high operating cost and financial burden of urban access regulation policies should also be emphasized [61]. As previously mentioned, implementing regulatory policies in isolation not only proves inefficient but also incurs significant costs. If a package of policy instruments is implemented cooperatively, the policy efficiency is much greater, but the cost of supporting facilities increases. For example, the LEZ aims to reduce vehicle emissions, and if people transition to low-emission vehicles (e.g., electric vehicles), there is an increased reliance on charging facilities, necessitating substantial infrastructure investments [110]. Simultaneously, investments should address social equity issues within the restricted area, such as increasing public transportation, bicycle, and pedestrian-related investments in low-income areas [16].

## 5. Discussion

In this study, the authors conducted a comprehensive review of the literature spanning the last five years (2019–2023) pertaining to urban access control policies to clarify the effectiveness of regulatory policies on sustainability and their own limitations. By examining numerous case studies from around the world and synthesizing the perspectives of other researchers, the authors categorized the findings according to the three pillars of sustainability: environment, society, and economics. It is noteworthy that there are overlapping aspects within these three pillars (see Figure 5), and the complexity increases with the overlapping levels.

The authors identified the shortcomings and challenges inherent in current urban access control policies. As Figure 6 illustrates, there are five potential issues that may arise during the implementation of urban access regulations, which can be termed "five paradoxes in urban access regulation".

The vision of urban access control is to reduce negative externalities from urban transportation and build sustainable transportation and more livable cities. However, these paradoxes stem from the shortcomings of the policies, which undermine the anticipated benefits, transform one problem into another dimension, or simply change the time and space in which the problem occurs, increasing social resistance to policy implementation and creating a vicious circle. These paradoxes can explain why urban access control policies are not supposed to be implemented in isolation or for long periods, as discussed in many scholarly works.

(1) The first issue is the emission reduction paradox, especially because many governments have established subsidies and incentives for renewable energy vehicles (some restriction exemptions, as mentioned above). An increasing number of vehicles (either fuel or electric vehicles) is likely to turn environmental issues into social issues (congestion). In addition, more vehicles offset some of the environmental benefits of restriction policies. For example, in areas with license plate number restrictions, people can circumvent such restrictions by purchasing a second cheap vehicle. An increase in the number of vehicles on roads also results in an increase in urban traffic noise;

(2) The essence of the spillover effect is the redistribution of traffic flows at the spatio-temporal level in congested or polluted areas rather than a reduction in private transport or a shift to low-carbon transport. These redistributed traffic flows predominantly occur at the boundaries of restricted areas and urban downtowns. Compared with city centers, residents residing near these boundaries typically have lower incomes and longer travel times, rendering the augmented traffic volume resulting from regulatory policies inequitable;

(3) The agglomeration effect arises when existing public transport or green transport services and infrastructure are insufficient to accommodate the significant influx of passengers within a short period as a result of the urban traffic regulation policy. As mentioned earlier, this often occurs at popular stations and locations. Shortage of passenger carrying capacity increases travel time and costs (e.g., queues due to overcrowding);

(4) From the society section, high-income groups have more pronounced car dependency, and they view private vehicle ownership as a status symbol. These individuals can afford the increased cost of private transportation due to restriction policies. They may even opt to purchase additional private vehicles or switch to electric vehicles to maintain private transport. In this case, urban access control policies inadvertently target middle and low-income populations, fostering resentment and resistance among these larger demographic groups, especially given their comparatively lower economic standing;

(5) Lastly, urban traffic regulations have limitations in the spatio-temporal dimension. On the one hand, access regulations cannot serve as long-term policies because the marginal benefits diminish with technological advancements and urbanization. On the other hand, it is important to take into account the industrial scale, climate, geography, and human and transportation circumstances of the city before implementing the policy instead of blindly duplicating others' regulatory models.

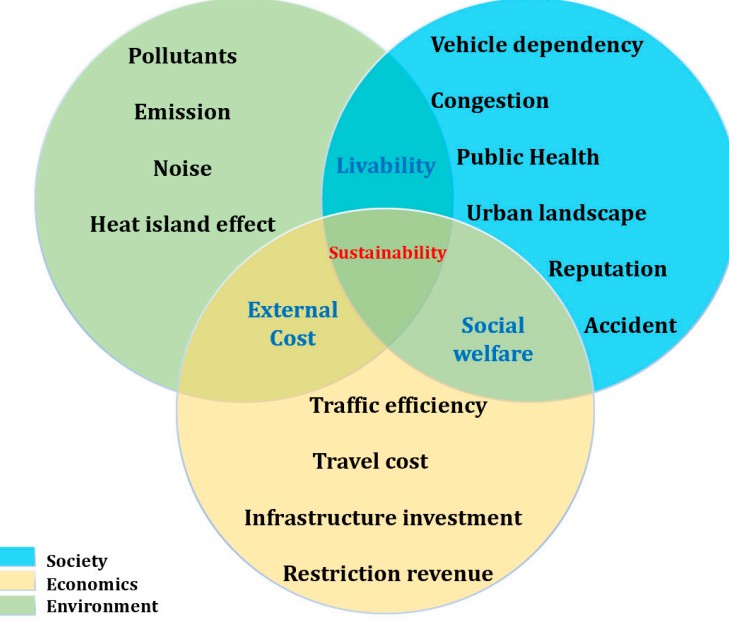

**Figure 5.** Summary of aiming predicaments by urban access control policies in this paper (source: own edition).

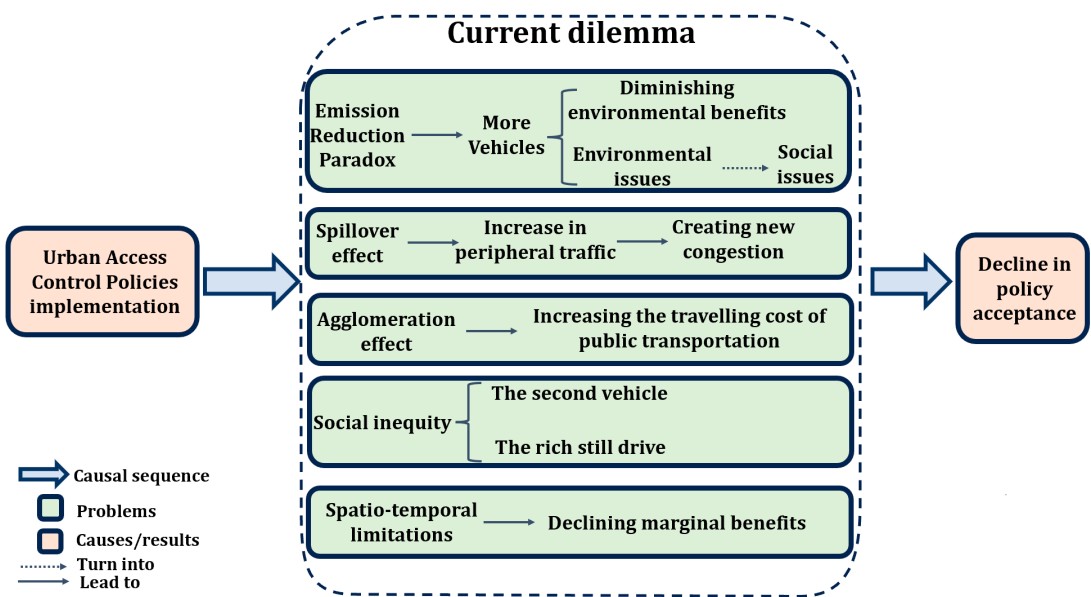

**Figure 6.** Potential problems while implementing urban access controls (source: own edition).

In addition to identifying the limitations, these studies highlighted some recognized approaches to enhance policy effectiveness. First, policy instruments implementation in silos should be avoided. Instead, a combination of policy instruments should be pursued to create synergies. These combinations can be intra-policy (a combination of various urban mobility restrictions), such as the simultaneous implementation of several of the restrictions discussed (depending on the local circumstances). It is easier for people to circumvent regulations in the case of a single policy, such as the spatio-temporal redistribution of traffic volume described earlier. The combinations can also be inter-policy (a combination of urban mobility restrictions and other policies), such as incentives and subsidies for public transport, vehicle purchase restrictions, etc. The purpose of such a scheme is to compensate low-income people who can only use public transport and green transport. Restrictions on private transportation could lead to increased public transportation ridership, potentially resulting in temporary discomfort and overcrowding for regular users. Those who give up private transport are likely to complain about the policies because of the lower quality of public transport. Vehicle purchase restriction policies can mitigate the emission reduction paradox by addressing issues such as the acquisition of secondary vehicles while also promoting social equity.

The second factor is stakeholders' involvement in policymaking. Shifting away from the traditional model of unilateral government policymaking towards a framework that embraces societal engagement would be advantageous. Engaging a diverse array of stakeholders, including government officials, experts, and citizen representatives, in the formulation of urban transportation regulation policies can help mitigate limitations and address blind spots from a broader perspective. This approach can help increase the public's acceptance of restriction policies and ensure that diverse viewpoints and insights are incorporated into the policymaking process.

## 6. Conclusions

The authors investigated the pathways to sustainability through urban access control policies within the context of urbanization. This study applied the SLR method to 89 studies sourced from three scientific databases, which included case studies and previous reviews. By analyzing previous research and ideas, the authors synthesized the three dimensions of sustainability (environment, society, and economics) to demonstrate strategies for enhancing the efficacy of urban access control policies while mitigating negative effects during implementation. Although it is difficult to completely split these policies into the three

pillars of sustainability, environmental-targeting policies are most widely implemented worldwide. For instance, both the license plate digit restriction and the LEZ aim to reduce emissions as well as air pollutants, one from the temporal dimension and the other from the spatial dimension. Regarding societal impact, residents residing near key infrastructure (e.g., bus stations, metro lines, major roads) and those disproportionately affected by transportation externalities (e.g., congested or high-density areas) exhibit stronger support for access regulation policies. Moreover, new technologies are capable of generating greater potential to alleviate the negative externality and produce more effective policies.

Most of the published studies targeted policy effectiveness and achievement, but ignoring that urban access control policies themselves might also introduce negative externalities. These five paradoxes discussed in this paper would warn policymakers about further potential failure factors and come up with improvements in existing urban access control policies.

However, there are some limitations to this study. First, while the authors aimed to explore a global perspective, the geographical distribution of case studies is not uniform, with a predominant concentration in Asia and Europe, as depicted in Figure 2. Second, as stated in the methodology section, research on policies is time-sensitive, which also applies to this study. Third, the case studies are discussed only regarding their results and shortcomings, without an in-depth study of the local context and circumstances, such as demographic and economic data, city scale, and other relevant factors.

Based on this paper's scientific contribution, future studies can conduct in-depth research on urban access regulations' externalities through accurate urban transport data and simulation. Transport authorities would be more cautious while considering the paradoxes. Furthermore, there is potential for in-depth investigations into research gaps, such as exploring the sensitivity between the quality of public transport and urban traffic regulations, as well as examining access control policies from the perspective of urban freight management. These avenues of inquiry could contribute to a more comprehensive understanding of the complexities surrounding urban access control policies and their implications for sustainable urban development.

**Supplementary Materials:** The following supporting information can be downloaded at: https://www.mdpi.com/article/10.3390/urbansci8020029/s1, Table S1: List of selected articles.

**Author Contributions:** Conceptualization, Y.M. and F.M.; methodology, Y.M. and F.M.; validation, F.M.; formal analysis, Y.M.; investigation, Y.M.; resources, Y.M.; data curation, Y.M.; writing—original draft preparation, Y.M.; writing—review and editing, F.M.; visualization, Y.M.; supervision, F.M.; project administration, F.M. All authors have read and agreed to the published version of the manuscript.

**Funding:** This research received no external funding.

**Data Availability Statement:** No new data were created.

**Acknowledgments:** This research was supported by OTKA—K20—134760—Heterogeneity in user preferences and its impact on transport project appraisal led by Adam TOROK. The authors also appreciate Gabriel Ogunkunbi 's enlightenment on the aspect of the literature search.

**Conflicts of Interest:** The authors declare no conflicts of interest.

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
