# Peer review of "Review of Urban Access Regulations from the Sustainability Viewpoint"

_urbansci, doi:10.3390/urbansci8020029_

Round 1

Reviewer 1 Report

Comments and Suggestions for Authors

Dear authors,

Thank you for the idea of conducting a literature review on such an important topic. Systematic reviews are highly useful for advancing scientific knowledge. However, I can only recommend major revisions due to four major problems. These problems can be relatively easily addressed by the authors of the submitted manuscript. 

Problem no. 1: Criteria listed in Table 2 are rather vague and does not allow for replicating the study:

  • Criteria 1 - “The title and abstract cover urban access regulation policies and address the three dimensions of sustainability” - do they have to address it simultaneously?;

  • Criteria 2 - Which is the list of reviewed papers? Which one has been selected at each filtering stage? And which ones have been added because they have been cited multiple times? What does multiple times mean? The full list of papers should be made public;

  • Criteria 3 - ok, clear;

  • Criteria 4 - ok, clear;

  • Criteria 5 - What does criteria “Literature lapsed or not included within the affiliations access rights” refer to? For me it is unclear. I cannot understand its meaning;

  • Criteria 6 - ok, clear;

Problem no 2: When it comes to presenting the findings of the literature review (sections 3.2, 3.3, 3.4), the authors engage in enumeration of the most interesting policies that have been approached in the scientific papers that they have included in the review. While such examples are useful, the authors should not limit themselves to delivering only such an enumeration, as it renders the manuscript hard to read, and to seem like a collage of ideas from the literature, instead of manuscript built around an idea developed by its authors.

Problem no 3: What is the main added value of the paper? I think it is a very good idea to conceptualize and highlight the 5 paradoxes, but more work is necessary to clarify them. What should one understand by “paradox” in urban access regulation? I would challenge the author to give a definition as it would make the discourse more clear.

Also, some minor aspects:

  1. Does “no. of literature” from table 1 refer to “no. of papers”? Does this number refer to the number of papers before or after filtering? Which level of filtering? (first, second, third, etc.)?

  2. Row 113 - “a lower density…” - what type of density? Population density?

  3. "In this paper, the specific problem categorization relied on the approach of 135

  4. Ogunkunbi and Meszaros [19].” A few words about their classification would be useful for those who are unfamiliar with their approach. Their approach seems very new and not well known in the literature.

  5. I would encourage the authors to give a definition for each of the three dimensions of urban mobility, as they have different meanings in the literature from one paper to another. What are the most frequent definitions that one can find in the literature? and what is the definition/meaning of the three dimensions that the authors of the submitted manuscript have worked with?

  6. Figure 4 is rather confusing because not all the items listed inside circles are „access control policies”. I think it could be either deleted or redesigned.

Finally I find that the typology of urban vehicle access regulations delivered in rows 120-124 is enlightening and highly useful. The same is true for the specific classification of urban access regulations  from table 3. Also, the identification and discussion around the „five” paradoxes is enlightening and could be one of the major contributions of the submitted manuscript.

Looking forward to a better version of the paper and to see your paper published.

Kind regards,

The reviewer

Reviewer 2 Report

Comments and Suggestions for Authors

In the present manuscript, the authors review the urban vehicle access control policies derived from absurd spatiotemporal dimensions that aim to eliminate the negative externalities of traffic caused by urbanization. However, I will comment on some aspects to improve the quality of the article, and the suggested changes should be highlighted.

- Authors must use the correct verb tense in each article section.

- The authors misuse acronyms. The correct way is to capitalise the first letter of the meaning of the acronym, i.e. "Electric Vehicle (EV)". This error must be corrected throughout the document.

- At the end of the Introduction Section, authors must include a summary of the sections the manuscript will cover.

- Authors must add the Related Works Section.

- What is the SLR methodology that you have used?

- For the search process for review articles, it is rare to use Google Scholar, but it is better to use digital libraries such as IEEE Xplore, ACM, MDPI, etc. It is necessary to reformulate the search because doing it through Google Scholar tends to repeat or find articles from low-quality journals.

- Some Figures have yet to be referenced.

- Authors must not use Phrasal Verbs.

- Could the articles include be in a language other than English?

- How many studies have you found in their entirety with the search strings? The authors must indicate how they have excluded articles until they reach 89 studies.

- It is necessary to improve the quality of the conclusions, and the conclusions cannot have references.

Comments on the Quality of English Language

In the present manuscript, the authors review the urban vehicle access control policies derived from absurd spatiotemporal dimensions that aim to eliminate the negative externalities of traffic caused by urbanization. However, I will comment on some aspects to improve the quality of the article, and the suggested changes should be highlighted.

- Authors must use the correct verb tense in each article section.

- The authors misuse acronyms. The correct way is to capitalise the first letter of the meaning of the acronym, i.e. "Electric Vehicle (EV)". This error must be corrected throughout the document.

- At the end of the Introduction Section, authors must include a summary of the sections the manuscript will cover.

- Authors must add the Related Works Section.

- What is the SLR methodology that you have used?

- For the search process for review articles, it is rare to use Google Scholar, but it is better to use digital libraries such as IEEE Xplore, ACM, MDPI, etc. It is necessary to reformulate the search because doing it through Google Scholar tends to repeat or find articles from low-quality journals.

- Some Figures have yet to be referenced.

- Authors must not use Phrasal Verbs.

- Could the articles include be in a language other than English?

- How many studies have you found in their entirety with the search strings? The authors must indicate how they have excluded articles until they reach 89 studies.

- It is necessary to improve the quality of the conclusions, and the conclusions cannot have references.

Round 2

Reviewer 1 Report

Comments and Suggestions for Authors

Congrats for the new and significantly improved version of the paper, especially in regard with methodological transparency and replicability. I find that the improvements are satisfactory, though not the best that I would have expected. Nevertheless I reccommend minor revisions, especially in regard to giving details or a definition for one unclear affirmation that is important: "Literature lapsed or not included within the affiliations access rights".

The paper would have contributed more to the litterature if more emphasis would have been put on the added value of the review, and less on enumerating most interesting results.

The list of reviewed papers should be giving as appendix or supplementary material.
